# Anisotropic Gaussian kernel adaptive filtering by Lie-group dictionary learning

**Tomoya Wada[1]\*, Kosuke Fukumori[1]\*, Toshihisa Tanaka[1]\*, Simone Fiori[2]\***

**1** Department of Electrical and Electronic Engineering, Tokyo University of Agriculture and Technology, Koganei-shi, Tokyo, Japan, **2** Università Politecnica delle Marche, Ancona, Italy

\* wada15@sip.tuat.ac.jp (TW); fukumori17@sip.tuat.ac.jp (KF); tanakat@cc.tuat.ac.jp (TT); s.fiori@univpm.it (SF)

**Data Availability Statement:** All relevant data are within the manuscript.

**Funding:** TT: JSPS KAKENHI Grant Number 17H01760 (URL: https://www.jsps.go.jp/english/e-grants/) SF: 2016 "Research in Pairs" program by

## Abstract

The present paper proposes a novel kernel adaptive filtering algorithm, where each Gaussian kernel is parameterized by a center vector and a symmetric positive definite (SPD) precision matrix, which is regarded as a generalization of scalar width parameter. In fact, different from conventional kernel adaptive systems, the proposed filter is structured as a superposition of non-isotropic Gaussian kernels, whose non-isotropy makes the filter more flexible. The adaptation algorithm will search for optimal parameters in a wider parameter space. This generalization brings the need of special treatment of parameters that have a geometric structure. In fact, the main contribution of this paper is to establish update rules for precision matrices on the Lie group of SPD matrices in order to ensure their symmetry and positive-definiteness. The parameters of this filter are adapted on the basis of a least-squares criterion to minimize the filtering error, together with an $\ell_1$-type regularization criterion to avoid overfitting and to prevent the increase of dimensionality of the dictionary. Experimental results confirm the validity of the proposed method.

## 1 Introduction

Adaptive filtering is a technique to update the parameters of a signal/data processing structure [1]. In this paper, we deal with kernel-based adaptive filtering. A kernel adaptive filter is a kind of nonlinear filter that exploits a kernel method, which is a technique to construct effective nonlinear systems [2]. Kernel adaptive filters found widespread applications in diverse fields, ranging from stock market prediction [3] to acoustic echo cancellation [4] and visual object tracking [5].

In kernel adaptive filtering, most kernels present the following form [6]:

$$\kappa(\cdot, c; \gamma) := \exp\left(-\gamma \| \cdot - c\|^2\right), \tag{1}$$

where parameters $c \in \mathbb{R}^L$ and $\gamma > 0$ represent the *center* and the *width* of a Gaussian kernel, respectively. In other words, this kind of kernel presents only two parameters, namely, mean and variance (also referred to as scalar precision).

the National Center for Theoretical Sciences (NCTS), Taiwan (URL: http://www.ncts.ntu.edu.tw/) The funders had no role in study design, data collection and analysis, decision to publish, or preparation of the manuscript.

**Competing interests:** The authors have declared that no competing interests exist.

## 1.1 Related work

Several instances of nonlinear adaptive filtering have been reported in the scientific literature. Among them, kernel adaptive filtering developed in a reproducing kernel Hilbert space (RKHS) is known as an efficient online nonlinear approximation approach [7, 8]. Well-known kernel adaptive filtering algorithms are kernel least mean square (KLMS) [9–12], kernel normalized least mean square (KNLMS), kernel affine projection algorithms (KAPA) [13, 14], and kernel recursive least squares (KRLS) [15]. In this context, it is worth citing fractional adaptive signal processing [16–20] as these modern filtering algorithms outperform their counterparts in terms of accuracy and convergence, for example in active noise control systems.

A distinguishing feature of kernel-based adaptive filtering is the ability to adjust the values of the parameters of each kernel so as to minimize the filtering error. To what concerns kernel centers, research endeavors suggested to move all the center vectors in the dictionary to minimize the squared filtering error [21–23]. To what concerns kernel widths, it is known that the widths of the kernels are important parameters that contribute to improve the performance of kernel machines [24–28] and some attempts to adaptively estimate the widths of the kernels have been reported [27, 28]. Moreover, in a recent work [6], Wada *et al.* have proposed an adaptive update method for both the Gaussian center and width concurrently. The above mentioned papers have addressed the problem of estimating a precision parameter of the Gaussian model given as in (1). However, this is a special case of multivariate Gaussian kernel function.

The structure of most kernel adaptive filtering algorithms grows linearly with each new input sample. A solution to cope with this problem is to build a *dictionary*. Well-known criteria for dictionary learning are novelty [29], approximate linear dependency (ALD) [15], surprise [30], and coherence-based criterion [31]. Another known criterion is $\ell_1$-regularization [32, 33], which sets some coefficients to zero and discards the corresponding entries. In this instance, a model dynamically changes, in that new members may be added to a dictionary and old members may be suppressed from a dictionary.

## 1.2 Innovative contribution

It should be noted that a kernel of the form (1) implicitly assumes uncorrelatedness between components in the sample vector, which implies that the kernel can be isotropic. However, observed samples usually present some sort of mutual correlation [34, 35].

In this paper, we employ a generalized Gaussian kernel defined as

$$\kappa(\cdot, c; \Gamma) := \exp\left(-(\cdot - c)^\top \Gamma(\cdot - c)\right), \tag{2}$$

where $\Gamma \in \mathbb{R}^{L \times L}$ is a symmetric positive definite (SPD) matrix. We refer to $\Gamma$ as a precision matrix, which has no constraint but positive definiteness, while the model of (1) can be regarded as a special case where $\Gamma = \gamma I$. In other words, in (1) the precision matrix is allowed to be only isotropic. Unlike (1), this general form has more degrees of freedom and therefore it is more flexible in modeling samples distributions; however, an adaptive method for finding the precision matrix is not straightforward. We will establish a dictionary learning method for generalized Gaussian kernel adaptive filtering. In a dictionary, each entry consists of a pair formed by a center vector and a precision matrix. The main contributions of the proposed method are (a) a model of the filter consisting of kernels with a different precision matrix each (b) an update rule for each center vector, and (c) a learning rule for each precision matrix on the SPD manifold in order to ensure their symmetry and positivity definiteness during adaptation.

### 1.3 Organization and list of abbreviations

Section 2 presents general concepts in kernel adaptive filtering. Section 3 proposes a dictionary learning method for the generalized Gaussian kernel adaptive filtering. Section 4 shows the results of numerical experiments to evaluate the efficacy of the proposed method. Section 5 concludes the paper. A list of abbreviations used within this paper is presented in Table 1.

## 2 Kernel adaptive filters

Kernel adaptive filters possess noteworthy features [8], such as universal approximation ability, absence of local minima and moderate complexity in terms of computation burden and memory. In this section, we first discuss sample distributions modeling in the context of kernel adaptive filtering and then we briefly review kernel adaptation algorithms.

In kernel adaptive filtering, an input sequence $\boldsymbol{u}^{(n)} \in \mathcal{U} \subset \mathbb{R}^L$ is mapped to a RKHS $(\mathcal{H}, \langle \cdot, \cdot \rangle)$ on $\mathcal{U}$ induced from a positive definite kernel $\kappa(\cdot, \cdot) : \mathcal{U} \times \mathcal{U} \to \mathbb{R}$. Here, symbol $\mathcal{U}$ denotes a multidimensional input space, while symbol $\langle \cdot, \cdot \rangle : \mathcal{H} \times \mathcal{H} \to \mathbb{R}$ denotes an inner product in the RKHS. A RKHS $\mathcal{H}$ can implicitly increase the dimensionality of a feature space that enables us to represent non-linear signals, which are generated by a non-linear system [36]. The short-term scalar output sequence of the filter is computed as

$$y^{(n)} = \langle \varphi(\boldsymbol{u}^{(n)}), P^{(n)} \rangle, \tag{3}$$

where $P^{(n)} \in \mathcal{H}$ denotes a filter weight vector at time $n$ and $\phi : \mathcal{U} \to \mathcal{H}$ denotes a nonlinear mapping. In general, the inner product in a high dimensional space is not given in an explicit form. Rather, the inner product in a RKHS can be calculated by using the properties of RKHS, namely: (i) all elements in a RKHS are constructed by a kernel $\kappa(\cdot, \cdot)$, (ii) it is convenient to choose $\varphi(\boldsymbol{u}) = \kappa(\cdot, \boldsymbol{u})$, (iii) it holds that $\langle \kappa(\cdot, \boldsymbol{u}_i), \kappa(\cdot, \boldsymbol{u}_j) \rangle = \kappa(\boldsymbol{u}_i, \boldsymbol{u}_j)$ [2, 31]. The Fig 1 shows a schematic of the adaptive filter.

We consider the problem of adaptively estimating the weights $P^{(n)}$. It is known [31] that $P^{(n)}$ can be written as

$$P^{(n)} = \sum_{j \in \mathcal{J}^{(n)}} h_j^{(n)} \kappa(\cdot, \mathbf{c}_j), \tag{4}$$

where the $h_j^{(n)} \in \mathbb{R}$ are scalar weight coefficient for $\kappa(\cdot, \mathbf{c}_j)$. Here, $\{\mathbf{c}_j\}_{j \in \mathcal{J}^{(n)}}$ is a set of input samples, termed *dictionary*. The symbol $\mathcal{J}^{(n)}$ denotes an index set of dictionary elements at time $n$.

**Table 1. List of abbreviations and their meaning.**

| Abbreviation | Explanation |
|---:|---|
| ALD | approximate linear dependency |
| KAPA | kernel affine projection algorithms |
| KLMS | kernel least mean square |
| KNLMS | kernel normalized least mean square |
| KRLS | kernel recursive least squares |
| LMS | least mean squares |
| MEG | matrix exponentiated gradient |
| MSE | mean squared error |
| NMEG | normalized matrix exponentiated gradient |
| RKHS | reproducing kernel Hilbert space |
| SPD | symmetric positive definite |

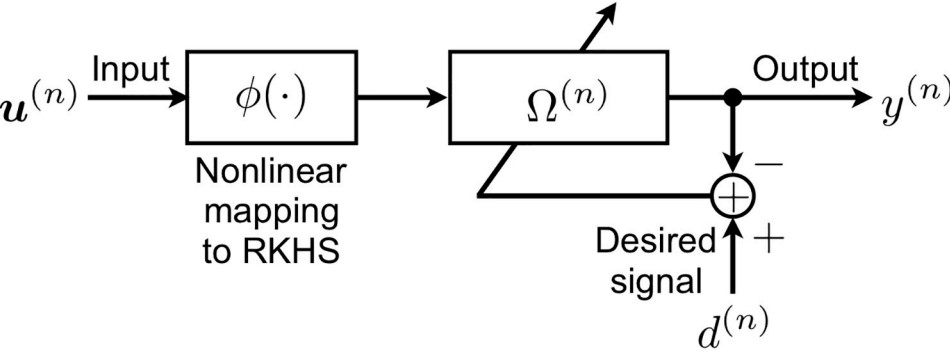

**Fig 1. Schematic of a RKHS adaptive filter.**

From Eqs (3) and (4), the filter output is written as

$$y^{(n)} = \sum_{j \in \mathcal{J}^{(n)}} h_j^{(n)} \kappa(\boldsymbol{u}^{(n)}, \boldsymbol{c}_j) = \boldsymbol{h}^{(n)\top} \boldsymbol{\kappa}^{(n)},$$

(5)

where

$$\boldsymbol{h}^{(n)} := [h_{j_1^{(n)}}^{(n)}, h_{j_2^{(n)}}^{(n)}, \ldots]^\top, \quad \boldsymbol{\kappa}^{(n)} := [\kappa(\boldsymbol{u}^{(n)}, \boldsymbol{c}_{j_1^{(n)}}), \kappa(\boldsymbol{u}^{(n)}, \boldsymbol{c}_{j_2^{(n)}}), \ldots]^\top.$$

(6)

Notice that, in the above equations, for the sake of notation conciseness the kernel functions have been indicated without reference to the kernel width parameter $\gamma$ that takes the same value in each kernel.

In kernel adaptive filtering algorithms, obsolete kernel functions cannot be discarded, which is a serious limitation of these algorithms, in particular in the presence of a non-stationary environment. In order to fix this issue, a dictionary may be constructed by means of $\ell_1$-regularization [32, 33] which promotes sparsity, hence improving the efficiency of the dictionary. Let $d^{(n)} \in \mathbb{R}$ denotes a desired output signal at time $n$. The cost function for the adaptive kernel algorithm is written as follows:

$$A^{(n)} := \left| d^{(n)} - \boldsymbol{h}^{(n)\top} \boldsymbol{\kappa}^{(n)} \right|^2 + \mu \underbrace{\sum_{j \in \mathcal{J}^{(n)}} w_j^{(n)} |h_j^{(n)}|}_{=: \beta^{(n)}},$$

(7)

where $\beta^{(n)}$ and $\mu$ play the role of a weighted $\ell_1$ norm and of a regularization parameter, respectively. Here, the weights $\{w_j^{(n)}\}_{j \in \mathcal{J}^{(n)}}$ are dynamically adjusted as $w_j^{(n)} = 1/(|h_j^{(n)}| + \rho)$ [33], with a small constant $\rho > 0$ to prevent the denominator from vanishing.

It is worth noticing that a conventional stochastic gradient descent method would be ineffective to seek the minimum of the cost function (7) since the weighted $\ell_1$ norm is not smooth. However, since the cost function $A^{(n)}$ is convex, a forward-backward splitting scheme [37] may be applied. A forward-backward splitting scheme reads:

$$\boldsymbol{h}^{(n+1)} = \text{prox}_{\lambda \mu \beta^{(n)}} \left[ \overline{\boldsymbol{h}^{(n)}} + \frac{\mu (d^{(n)} - \overline{\boldsymbol{h}^{(n)\top}} \overline{\kappa^{(n)}}) \overline{\kappa^{(n)}}}{\sigma + \|\overline{\kappa^{(n)}}\|^2} \right],$$

(8)

where $\overline{\boldsymbol{h}^{(n)}} := [\boldsymbol{h}^{(n)\top}, 0]^\top$, $\overline{\kappa^{(n)}} := [\boldsymbol{\kappa}^{(n)\top}, \kappa(\boldsymbol{u}^{(n)}, \boldsymbol{u}^{(n)})]^\top$, the coefficient $\lambda > 0$ denotes a step size, the coefficient $\sigma$ denotes a stabilization parameter, and $\|\cdot\|$ denotes a standard vector 2-norm.

The symbol 'prox' denotes the proximal operator [37], which is defined as follows: given a vector $\boldsymbol{q} := [q_1, q_2, \ldots, q_r]^\top \in \mathbb{R}^r$, it holds that

$$(\text{prox}_{\lambda\mu\beta^{(n)}}(\boldsymbol{\alpha}))_j := \text{sgn}\{q_j\} \max\{|q_j| - \lambda\mu w_j^{(n)}, 0\}. \tag{9}$$

For further details on this technique, interested readers might consult [6, 38, 39]. This learning rule promotes the sparsity of the $h_j^{(n)}$'s, which results in some coefficient $h_j^{(n)}$ approaching zero and the corresponding center vector $\mathbf{c}_j$ getting removed from the dictionary.

## 3 Model and dictionary learning for generalized Gaussian kernel adaptive filtering

As recalled in the introduction, most kernel machines using Gaussian kernel functions implicitly assume uncorrelatedness within the sample-variables, even though observed samples usually present correlation. In the following, a flexible filtering structure based on a superposition of generalized Gaussian functions is proposed and algorithms for learning its parameters are established.

### 3.1 Adaptive kernel filter based on a superposition of generalized Gaussian functions

The proposed model is structured as a superposition of generalized Gaussian kernels given as in (2) with time-varying centers $\boldsymbol{c}_j^{(n)}$ and precision matrices $\Gamma_j^{(n)}$. The output sequence of one such kernel adaptive filter is computed as

$$
\begin{aligned}
y^{(n)} &= \sum_{j \in \mathcal{J}^{(n)}} h_j^{(n)} \kappa(\boldsymbol{u}^{(n)}, \boldsymbol{c}_j^{(n)}; \Gamma_j^{(n)}) \\
&= \sum_{j \in \mathcal{J}^{(n)}} h_j^{(n)} \exp\left(-(\boldsymbol{u}^{(n)} - \boldsymbol{c}_j^{(n)})^\top \Gamma_j^{(n)} (\boldsymbol{u}^{(n)} - \boldsymbol{c}_j^{(n)})\right).
\end{aligned}
\tag{10}
$$

The corresponding dictionary at time $n$ is described by

$$\mathcal{D}^{(n)} := \{(\boldsymbol{c}_{j_1}^{(n)}, \Gamma_{j_1}^{(n)}), (\boldsymbol{c}_{j_2}^{(n)}, \Gamma_{j_2}^{(n)}), \ldots, (\boldsymbol{c}_{j_{r(n)}}^{(n)}, \Gamma_{j_{r(n)}}^{(n)})\}. \tag{11}$$

For the sake of completeness, let us discuss how a multikernel adaptive filter fits within the general theory of RKHS. An extended discussion on multikernel adaptive filtering may be found in [38, 40].

Let $\mathcal{H}_1$ and $\mathcal{H}_2$ denote two reproducing kernel Hilbert spaces and let $H := \mathcal{H}_1 \oplus \mathcal{H}_2$ denote their direct sum. The norm of the direct sum of $f_1 \in \mathcal{H}_1$ and $f_2 \in \mathcal{H}_2$, $f = (f_1, f_2) \in H$, is represented as [2]:

$$\|f\|_H^2 := \|f_1\|_{\mathcal{H}_1}^2 + \|f_2\|_{\mathcal{H}_2}^2. \tag{12}$$

In particular, if the two Hilber spaces are non-overlapping, namely $\mathcal{H}_1 \cap \mathcal{H}_2 = \{0\}$, the sum space $\mathcal{H} := \{f = f_1 + f_2 \mid f_1 \in \mathcal{H}_1, f_2 \in \mathcal{H}_2\}$ has the same structure of the space $H$ [2]. Consequently, the norm in $\mathcal{H}$ may be defined as:

$$\|f\|_{\mathcal{H}}^2 := \|f_1\|_{\mathcal{H}_1}^2 + \|f_2\|_{\mathcal{H}_2}^2. \tag{13}$$

Also, take a kernel $\kappa_1 \in \mathcal{H}_1$ and a kernel $\kappa_2 \in \mathcal{H}_2$. An element $f \in \mathcal{H}$ can be evaluated by the sum kernel $\kappa := \kappa_1 + \kappa_2$ [2]:

$$f(\boldsymbol{u}) = \langle f, \kappa(\cdot, \boldsymbol{u}) \rangle_{\mathcal{H}} = \langle f_1, \kappa_1(\cdot, \boldsymbol{u}) \rangle_{\mathcal{H}_1} + \langle f_2, \kappa_2(\cdot, \boldsymbol{u}) \rangle_{\mathcal{H}_2}. \tag{14}$$

The above construction may be generalized to an arbitrary number of Hilbert spaces without difficulty.

Assume now that $M$ different kernels $\{\kappa_m(\cdot, \cdot)\}_{m=1}^{M}$ are available. Denote by $\mathcal{H}_m$ a RKHS determined by the $m$-th kernel and define $\mathcal{H}$ as the corresponding sum space. In analogy to the simpler case (14), the output of the filter is obtained by combining a weight $P \in \mathcal{H}$ and the 'sum kernel' $\kappa \in \mathcal{H}$ as

$$y^{(n)} = \langle P, \kappa(\cdot, \boldsymbol{u}^{(n)}) \rangle_{\mathcal{H}} = \sum_{m=1}^{M} \langle P_m, \kappa_m(\cdot, \boldsymbol{u}^{(n)}) \rangle_{\mathcal{H}_m}, \tag{15}$$

where each weight $P_m \in \mathcal{H}_m$ and $P$ is identified with the (direct) sum of the single weights $P_m$. Since there is no need for the index set of the dictionary in each RKHS to equate each other [38], the filter structure (10) may be identified as a multikernel adaptive filter with time-varying weights:

$$y^{(n)} = \langle P^{(n)}, \kappa(\cdot, \boldsymbol{u}^{(n)}) \rangle_{\mathcal{H}} = \sum_{j \in \mathcal{J}^{(n)}} \langle P_j^{(n)}, \kappa(\cdot, \boldsymbol{u}^{(n)}; \Gamma_j^{(n)}) \rangle_{\mathcal{H}_j}, \tag{16}$$

with the convention that $P_j^{(n)} := h_j^{(n)} \kappa(\cdot, \boldsymbol{c}_j^{(n)}; \Gamma_j^{(n)})$.

## 3.2 Center vectors adaptation

In this subsection, a dictionary learning method for generalized Gaussian kernel adaptive filtering is proposed. To update the center vectors, we chose the loss function:

$$\begin{aligned} F^{(n)}(\mathcal{D}^{(n)}) \quad &:= |e^{(n)}|^2 = |d^{(n)} - y^{(n)}|^2 \\ &= \left| d^{(n)} - \sum_{j \in \mathcal{J}^{(n)}} h_j^{(n)} \exp\left(-(\boldsymbol{u}^{(n)} - \boldsymbol{c}_j)^{\top} \Gamma_j^{(n)} (\boldsymbol{u}^{(n)} - \boldsymbol{c}_j)\right) \right|^2. \end{aligned} \tag{17}$$

Such criterion is a function of dictionary elements, namely, of center vectors as well as of precision matrices.

The adaptation of each center vector may be achieved by a gradient steepest descent algorithm:

$$\boldsymbol{c}_j^{(n+1)} = \boldsymbol{c}_j^{(n)} - \eta_c \left. \frac{\partial F^{(n)}(\boldsymbol{c}_j)}{\partial \boldsymbol{c}_j} \right|_{\boldsymbol{c}_j = \boldsymbol{c}_j^{(n)}} \tag{18}$$

where $\eta_c > 0$ denotes a step size and

$$\left. \frac{\partial F^{(n)}(\boldsymbol{c}_j)}{\partial \boldsymbol{c}_j} \right|_{\boldsymbol{c}_j = \boldsymbol{c}_j^{(n)}} = -4e^{(n)} h_j^{(n)} \kappa\left(\boldsymbol{u}^{(n)}, \boldsymbol{c}_j^{(n)}; \Gamma_j^{(n)}\right) \Gamma_j^{(n)} \left(\boldsymbol{u}^{(n)} - \boldsymbol{c}_j^{(n)}\right) \tag{19}$$

Let us remark that adaptation rules to move center vectors for the standard Gaussian kernel adaptive filters were also proposed in [21–23].

### 3.3 Precision matrices adaptation

In order to update the precision matrices, we consider two types of data-driven adaptation methods. One consists in applying the update rule for SPD matrices proposed in [41]. The other is a novel update rule where an effective normalization is employed. The Fig 2 illustrates, in a schematic way, these update rules. In order to update precision matrices, the same loss function (17) may be invoked.

**3.3.1 Matrix Exponentiated Gradient (MEG) adaptation.** To update the precision matrices in a dictionary $\{\Gamma_j\}_{j \in \mathcal{J}^{(n)}}$ while preserving their SPD structure, a matrix exponentiated gradient (MEG) update [41] may be applied. The update rule for $\Gamma_j$ can be derived to minimize the loss function in (17):

$$\Gamma_j^{(n+1)} = \exp\left(\log \Gamma_j^{(n)} - \eta_{\mathrm{w}} \ \mathrm{sym}\left(\left.\frac{\partial F^{(n)}(\Gamma_j)}{\partial \Gamma_j}\right|_{\Gamma_j=\Gamma_j^{(n)}}\right)\right), \tag{20}$$

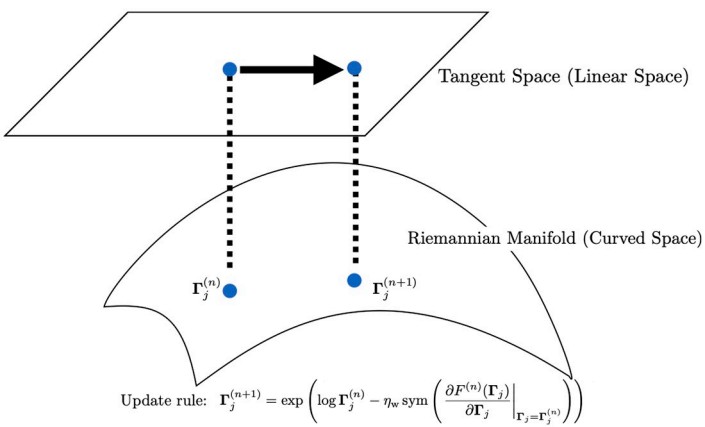

**(a)** Matrix exponentiated gradient (MEG) update. This rule requires the computation of $\log \boldsymbol{\Gamma}$ that can be unstable when the eigenvalues of $\boldsymbol{\Gamma}$ are very small.

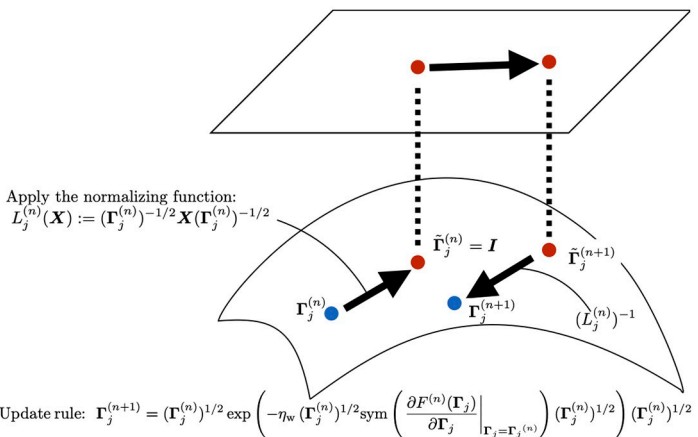

**(b)** Normalized MEG (NMEG) update. This rule can avoid the problem of MEG by using normalization.

**Fig 2. Conceptual diagrams of (a) MEG and (b) NMEG.**

where $\eta_w > 0$ denotes a step size and

$$\frac{\partial F^{(n)}(\Gamma_j)}{\partial \Gamma_j}\bigg|_{\Gamma_j = \Gamma_j^{(n)}} = \Gamma_j^{(n)} = 2e^{(n)}h_j^{(n)}\kappa(\boldsymbol{u}^{(n)}, \boldsymbol{c}_j^{(n)}; \Gamma_j^{(n)})(\boldsymbol{u}^{(n)} - \boldsymbol{c}_j)(\boldsymbol{u}^{(n)} - \boldsymbol{c}_j)^{\top}.$$

For a square matrix $\mathbf{X}$, $\mathrm{sym}(\boldsymbol{X}) := \frac{1}{2}(\boldsymbol{X} + \boldsymbol{X}^{\top})$ denotes the symmetric part of $\mathbf{X}$, while exp $(\mathbf{X})$ and log$(\mathbf{X})$ denote matrix exponential and principal matrix logarithm, respectively [41].

It is interesting to observe that the adaptation rule (20) may be re-interpreted in the framework of manifold calculus. In fact, let us define a retraction [42] as $R_{\mathbf{X}}(\mathbf{V}) := \exp(\log\mathbf{X} + \mathbf{V})$ in the SPD space, where $\mathbf{X}$ denotes any positive-definite symmetric matrix and $\mathbf{V}$ denotes any symmetric matrix of the same size. Also, let us define $\nabla_{\boldsymbol{X}}F := \mathrm{sym}\left(\frac{\partial F}{\partial \boldsymbol{X}}\right)$ as the Riemannian gradient of a loss function $F$ with respect to the SPD matrix $\mathbf{X}$ [43]. Then, the adaptation rule (20) may be re-framed as $\Gamma_j^{(n+1)} = R_{\Gamma_j^{(n)}}(-\eta_w\nabla_{\Gamma_j^{(n)}}F^{(n)})$.

**3.3.2 Normalized Matrix Exponentiated Gradient (NMEG) adaptation.**   Even though matrix exponentiated gradient updates each precision matrix $\Gamma$ while preserving its SPD structure, the computation of $\log\Gamma$ can be unstable when the eigenvalues of $\Gamma$ are too close to zero. A symmetric positive-definite matrix $\Gamma$ with $L$ all-distinct eigenvalues may be decomposed as $\boldsymbol{W}\,diag(\lambda_1, \lambda_2, \ldots, \lambda_L)\boldsymbol{W}^{\top}$, with $\mathbf{W}$ orthogonal. Therefore, $\log\Gamma = \boldsymbol{W}\,diag(\log\lambda_1, \log\lambda_2, \ldots, \log\lambda_L)\boldsymbol{W}^{\top}$: If an eigenvalue lays too close to zero, matrix logarithm becomes numerically unstable. In general, a matrix logarithm is well-defined only in a neighbor of the identity matrix $\mathbf{I}$. To overcome this problem, the following normalizing function by the current value $\Gamma_j^{(n)}$ is proposed:

$$L_j^{(n)}(\boldsymbol{X}) := (\Gamma_j^{(n)})^{-1/2}\boldsymbol{X}(\Gamma_j^{(n)})^{-1/2}, \tag{21}$$

where $X$ denotes any symmetric positive-definite matrix and $(\cdot)^{-1/2}$ denotes a combination of matrix inversion and symmetric square-rooting. On the basis of the observations recalled in the footnote [1], the inverse symmetric square root of a SPD matrix $\Gamma$ may be computed rather inexpensively by $\boldsymbol{W}diag(\lambda_1^{-1/2}, \lambda_2^{-1/2}, \ldots, \lambda_L^{-1/2})\boldsymbol{W}^{\top}$. Since a precision matrix $\Gamma_j^{(n)}$ is symmetric and positive-definite, its inverse always exists and its matrix square root always returns a symmetric, real-valued matrix. Let us remark how the introduced normalization keeps both symmetry and positive-definiteness of its argument, in fact, to what concerns symmetry:

$$\begin{aligned}(L_j^{(n)}(\boldsymbol{X}))^{\top} &= ((\Gamma_j^{(n)})^{-1/2}\boldsymbol{X}(\Gamma_j^{(n)})^{-1/2})^{\top} = (\Gamma_j^{(n)})^{-\top/2}\boldsymbol{X}^{\top}(\Gamma_j^{(n)})^{-\top/2} \\ &= (\Gamma_j^{(n)})^{-1/2}\boldsymbol{X}(\Gamma_j^{(n)})^{-1/2} = L_j^{(n)}(\boldsymbol{X}),\end{aligned} \tag{22}$$

and, to what concerns positive-definiteness:

$$\det(L_j^{(n)}(\boldsymbol{X})) = \det{}^2((\Gamma_j^{(n)})^{-1/2})\det(\boldsymbol{X}) = \det(\boldsymbol{X})/\det(\Gamma_j^{(n)}) > 0. \tag{23}$$

The inverse (de-normalizing) function associated to (21) reads:

$$(L_j^{(n)})^{-1}(\boldsymbol{X}) := (\Gamma_j^{(n)})^{1/2}\boldsymbol{X}(\Gamma_j^{(n)})^{1/2}. \tag{24}$$

Define a precision matrix $\Gamma$ normalized by (21) as $\tilde{\Gamma} := L_j^{(n)}(\Gamma)$. If we apply the MEG update to $\tilde{\Gamma}_j^{(n)}$ instead of $\Gamma_j^{(n)}$, we get the adaptation rule

$$\tilde{\Gamma}_j^{(n+1)} = \exp\left(\log\tilde{\Gamma}_j^{(n)} - \eta_w\mathrm{sym}\left(\frac{\partial F^{(n)}(\Gamma_j)}{\partial\tilde{\Gamma}_j}\bigg|_{\tilde{\Gamma}_j = \tilde{\Gamma}_j^{(n)}}\right)\right), \tag{25}$$

where $\Gamma_j = \Gamma_j(\tilde{\Gamma}_j)$ is to be thought of as a compound function, in fact, it holds that $\Gamma_j(\tilde{\Gamma}_j) := (L_j^{(n)})^{-1}(\tilde{\Gamma}_j)$. Notice that $\tilde{\Gamma}_j^{(n)}$ can be written as

$$\tilde{\Gamma}_j^{(n)} = L_j^{(n)}(\Gamma_j^{(n)}) = (\Gamma_j^{(n)})^{-1/2}\Gamma_j^{(n)}(\Gamma_j^{(n)})^{-1/2} = \boldsymbol{I}, \tag{26}$$

where $\boldsymbol{I} \in \mathbb{R}^{L \times L}$ is an identity matrix. Since $\log \boldsymbol{I} = \boldsymbol{0}$, the adaptation rule (25) simplifies to

$$\tilde{\Gamma}_j^{(n+1)} = \exp\left(-\eta_w \mathrm{sym}\left(\frac{\partial F^{(n)}(\Gamma_j(\tilde{\Gamma}_j))}{\partial \tilde{\Gamma}_j}\bigg|_{\tilde{\Gamma}_j = \tilde{\Gamma}_j^{(n)}}\right)\right), \tag{27}$$

To find the derivative of function $F^{(n)}(\Gamma_j)$ with respect to $\tilde{\Gamma}_j$, the following chain rule [44] is used:

$$\begin{aligned}\left(\frac{\partial F^{(n)}(\Gamma_j(\tilde{\Gamma}_j))}{\partial \tilde{\Gamma}_j}\right)_{kl} &= \mathrm{Tr}\left[\left(\frac{\partial F^{(n)}(\Gamma_j)}{\partial \Gamma_j}\right)^\top \frac{\partial \Gamma_j}{\partial(\tilde{\Gamma}_j)_{kl}}\right]\\ &= \mathrm{Tr}\left[\left(\frac{\partial F^{(n)}(\Gamma_j)}{\partial \Gamma_j}\right)^\top (\Gamma_j^{(n)})^{1/2}\frac{\partial \tilde{\Gamma}_j}{\partial(\tilde{\Gamma}_j)_{kl}}(\Gamma_j^{(n)})^{1/2}\right]\\ &= \mathrm{Tr}\left[\left(\frac{\partial F^{(n)}(\Gamma_j)}{\partial \Gamma_j}\right)^\top (\Gamma_j^{(n)})^{1/2}\boldsymbol{S}_{kl}(\Gamma_j^{(n)})^{1/2}\right]\\ &= \left((\Gamma_j^{(n)})^{1/2}\left(\frac{\partial F^{(n)}(\Gamma_j)}{\partial \Gamma_j}\right)^\top (\Gamma_j^{(n)})^{1/2}\right)_{lk},\end{aligned} \tag{28}$$

where the notation $(\mathbf{X})_{kl}$ indicates the $(k, l)$-th entry of matrix $\mathbf{X}$, $\mathrm{Tr}(\cdot)$ denotes matrix trace, and $\mathbf{S}_{kl}$ is the single-entry matrix [44], whose $(k, l)$-th entry is 1 and each other entry takes the value 0. From the property (28), we get

$$\begin{aligned}\frac{\partial F^{(n)}(\Gamma_j(\tilde{\Gamma}_j))}{\partial \tilde{\Gamma}_j} &= \left((\Gamma_j^{(n)})^{1/2}\left(\frac{\partial F^{(n)}(\Gamma_j)}{\partial \Gamma_j}\right)^\top (\Gamma_j^{(n)})^{1/2}\right)^\top\\ &= (\Gamma_j^{(n)})^{1/2}\frac{\partial F^{(n)}(\Gamma_j)}{\partial \Gamma_j}(\Gamma_j^{(n)})^{1/2},\end{aligned} \tag{29}$$

thanks to the symmetry of the involved matrices and expressions. Using the formula (29), the adaptation rule (27) can be written as

$$\begin{aligned}\tilde{\Gamma}_j^{(n+1)} &= \exp\left(-\eta_w \mathrm{sym}\left((L_j^{(n)})^{-1}\left(\frac{\partial F^{(n)}(\Gamma_j)}{\partial \Gamma_j}\bigg|_{\Gamma_j = \Gamma_j^{(n)}}\right)\right)\right)\\ &= \exp\left(-\eta_w (L_j^{(n)})^{-1}\left(\mathrm{sym}\left(\frac{\partial F^{(n)}(\Gamma_j)}{\partial \Gamma_j}\bigg|_{\Gamma_j = \Gamma_j^{(n)}}\right)\right)\right).\end{aligned} \tag{30}$$

Thanks to the normalizing function, we can update the precision matrices stably. Then, the $(n + 1)$-th precision matrix is obtained by applying the inverse normalizing function.

Therefore, the following update rule is derived:

$$
\begin{aligned}
\Gamma_j^{(n+1)} \quad & = (L_j^{(n)})^{-1}(\tilde{\Gamma}_j^{(n+1)}) = (\Gamma_j^{(n)})^{1/2}\tilde{\Gamma}_j^{(n+1)}(\Gamma_j^{(n)})^{1/2} \\
& = (\Gamma_j^{(n)})^{1/2}\exp\left(-\eta_{\mathrm{w}}\,(\Gamma_j^{(n)})^{1/2}\mathrm{sym}\left(\left.\frac{\partial F^{(n)}(\Gamma_j)}{\partial \Gamma_j}\right|_{\Gamma_j=\Gamma_j^{(n)}}\right)(\Gamma_j^{(n)})^{1/2}\right)(\Gamma_j^{(n)})^{1/2}.
\end{aligned}
\tag{31}
$$

From (31), we can see that unlike (20), this adaptation rule dose not require the computation of $\log\Gamma$. We call this adaptation rule normalized matrix exponentiated gradient (NMEG).

As a special instance, let us consider the case $L = 1$. The NMEG update rule in the case of $L = 1$ can be derived by replacing each precision matrix $\Gamma$ with a scalar parameter $\gamma > 0$ in (31):

$$
\gamma_j^{(n+1)} = \gamma_j^{(n)}\exp\left(-\eta_{\mathrm{w}}\gamma_j^{(n)}\left.\frac{\partial F^{(n)}(\gamma_j)}{\partial\gamma_j}\right|_{\gamma_j=\gamma_j^{(n)}}\right),
\tag{32}
$$

which apparently keeps each parameter $\gamma_j$ in the positive half-line during adaptation. The partial derivative of the cost function, in this case, reads

$$
\left.\frac{\partial F^{(n)}(\gamma_j)}{\partial\gamma_j}\right|_{\gamma_j=\gamma_j^{(n)}} = -2\gamma_j^{(n)}e^{(n)}h_j^{(n)}\kappa(\boldsymbol{u}^{(n)},\boldsymbol{c}_j^{(n)};\gamma_j^{(n)})\|\boldsymbol{u}^{(n)}-\boldsymbol{c}_j^{(n)}\|^2.
\tag{33}
$$

Such special case was proposed and discussed in the contributions [6, 28].

The adaptation rule (31) was derived on the basis of matrix normalization, therefore, it is legitimate to wonder if it constitutes a valid algorithm to update a matrix in the space of SPD tensors. The answer is positive, indeed, since the rule (31) may be regarded as an application of a general geodesic-based stepping rule on the manifold of symmetric positive-definite matrices endowed with the canonical metric, namely

$$
\Gamma_j^{(n+1)} = g_{\Gamma_j^{(n)}}\left(-\eta_{\mathrm{w}}\,\Gamma_j^{(n)}\mathrm{sym}\left(\left.\frac{\partial F^{(n)}(\Gamma_j)}{\partial\Gamma_j}\right|_{\Gamma_j=\Gamma_j^{(n)}}\right)\Gamma_j^{(n)}\right),
\tag{34}
$$

where the function $g_X(V)$ denotes a geodesic arc in the SPD space departing from a point $X$ in the direction $V$ and is given by

$$
g_X(V) := X^{1/2}\exp\left(X^{-1/2}VX^{-1/2}\right)X^{1/2},
\tag{35}
$$

as explained, for example, in [43] and [45]. Notice, in addition, that the argument of the function $g$ in (34) is proportional to the Riemannian gradient of the criterion function $F$ with respect to the canonical metric, as defined in the previous Subsection 3.3.1.

### 3.4 Sparse KNLMS incorporated with generalized Gaussian kernel parameters

To avoid overfitting and to prevent monotonic growth of a dictionary, the proposed adaptation rules for the generalized Gaussian parameters are applied jointly with an $\ell_1$-regularization [33]. The proposed method is summarized in Algorithm 1.

**Algorithm 1** Dictionary Learning for Generalized Gaussian Kernel Adaptive Filtering

```
1: Set precision matrices of kernels Γ_init.
2: Set the initial center vector c^(0) ← u^(0)
3: Add (c^(0), Γ_init) into the dictionary as the 1st member,
     D^(0) ← {(c^(0), Γ_init)}.
```

```
 4: for n > 1 do
 5:    Set the n-th center vector c⁽ⁿ⁾ ← u⁽ⁿ⁾
 6:    Add (c⁽ⁿ⁾, Γ_init) to the dictionary as a new member,
       𝒟⁽ⁿ⁾ ← 𝒟⁽ⁿ⁻¹⁾ ∪ {(c⁽ⁿ⁾, Γ_init)}.
 7:    for j ← 0 to size of 𝒟⁽ⁿ⁾ − 1 do
 8:       Update the center vectors c_j⁽ⁿ⁾ using (18).
 9:       Update the precision matrices Γ_j⁽ⁿ⁾ using MEG (20) or NMEG (31).
10:       Update the filter coefficients h_j according to a forward-back-
          ward splitting scheme (8).
11:    end for
12:    for j such that h_j = 0 do
13:       Remove the j-th element from the dictionary 𝒟⁽ⁿ⁾.
14:    end for
15:    n ← n + 1
16: end for
```

## 4 Numerical experiments

In this section, we compare the KNLMS-$\ell_1$ [33], the NMEG ($L = 1$) [6, 28] in (32), the MEG in (20), and the NMEG in (31) through three types of simulations. The first simulation is a time series prediction in a toy model defined by Gaussian functions with scalar widths. The second simulation is an online prediction in a toy model defined by Gaussian functions with precision matrices. The last simulation consists in an online prediction of the state of a Lorenz chaotic system. In these simulations, mean squared error (MSE) and mean dictionary size were adopted as the evaluation criteria. Both indices were averaged over 200 independent trials to compensate for statistical fluctuations in each single trial.

### 4.1 Time series prediction in a toy signal model constructed by standard Gaussian functions

Consider the following synthetic signal model:

$$d^{(n)} := 10 \exp\left(-5\|\boldsymbol{u}^{(n)} - [3,3]^\top\|^2\right) + 10 \exp\left(-0.2\|\boldsymbol{u}^{(n)} - [7,7]^\top\|^2\right), \tag{36}$$

corrupted by an additive zero-mean white Gaussian noise with standard deviation equal to 0.3. The input samples $\boldsymbol{u}^{(n)}$ are drawn from a 2-dimensional uniform distribution with support $[0, 10] \times [0, 10]$. The parameters values for the learning schemes utilized in this experiment are given in Table 2. In addition, the parameters values for the forward-backward splitting scheme are $\lambda = 0.09$ and $\sigma = 0.03$.

Figs 3 and 4 show the mean squared error and mean dictionary size of filters at each iteration, respectively. In Fig 3, the NMEG ($L = 1$), the MEG, and the NMEG show lower MSE than the KNLMS-$\ell_1$. This confirms the efficacy of updating the scalar widths $\gamma$ and the precision

**Table 2. Values of learning parameters in the experiment described in Subsection 4.1.**

| Learning algorithm | Parameters values |
|---|---|
| KNLMS-$\ell_1$ | $\gamma = 1.0$, $\mu = 1.0 \times 10^{-3}$, $\rho = 0.1$ |
| NMEG ($L = 1$) | $\gamma_{\text{init.}} = 1.0$, $\mu = 1.0 \times 10^{-3}$ <br> $\rho = 0.1$, $\eta_c = 1.0 \times 10^{-3}$, $\eta_w = 0.05$ |
| MEG | $\Gamma_{\text{init}} = \boldsymbol{I}$, $\mu = 1.0 \times 10^{-3}$ <br> $\rho = 0.1$, $\eta_c = 1.0 \times 10^{-3}$, $\eta_w = 0.05$, $L = 2$ |
| NMEG | $\Gamma_{\text{init}} = \boldsymbol{I}$, $\mu = 1.0 \times 10^{-3}$ <br> $\rho = 0.1$, $\eta_c = 1.0 \times 10^{-3}$, $\eta_w = 0.05$, $L = 2$ |

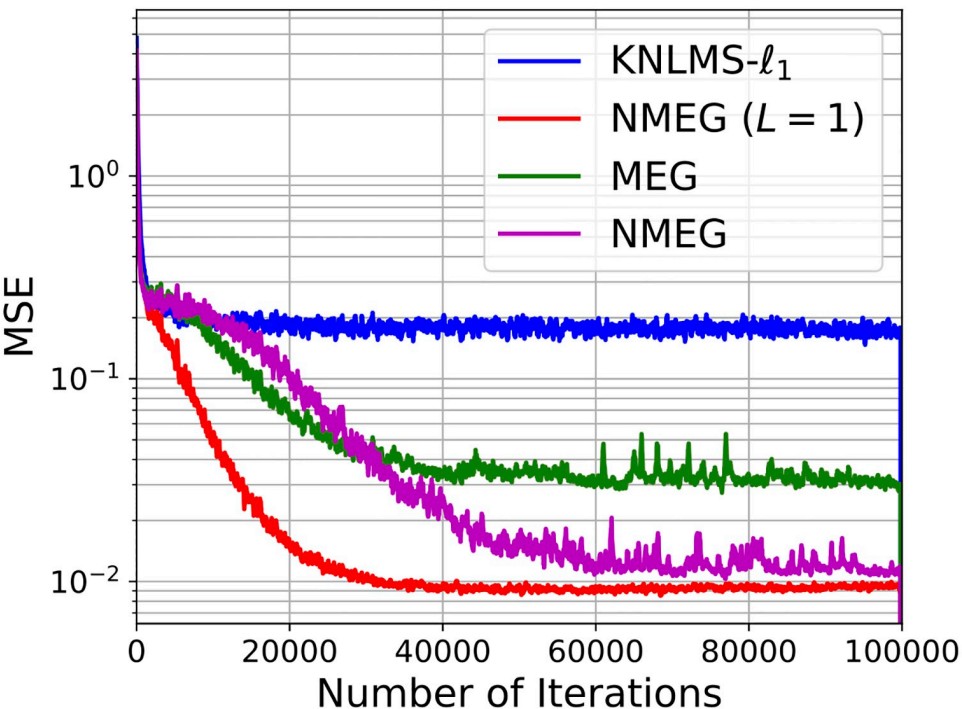

**Fig 3. Convergence curves of filters in the experiment described in the Section 4.1.** These results were obtained as averages over 200 independent trials.

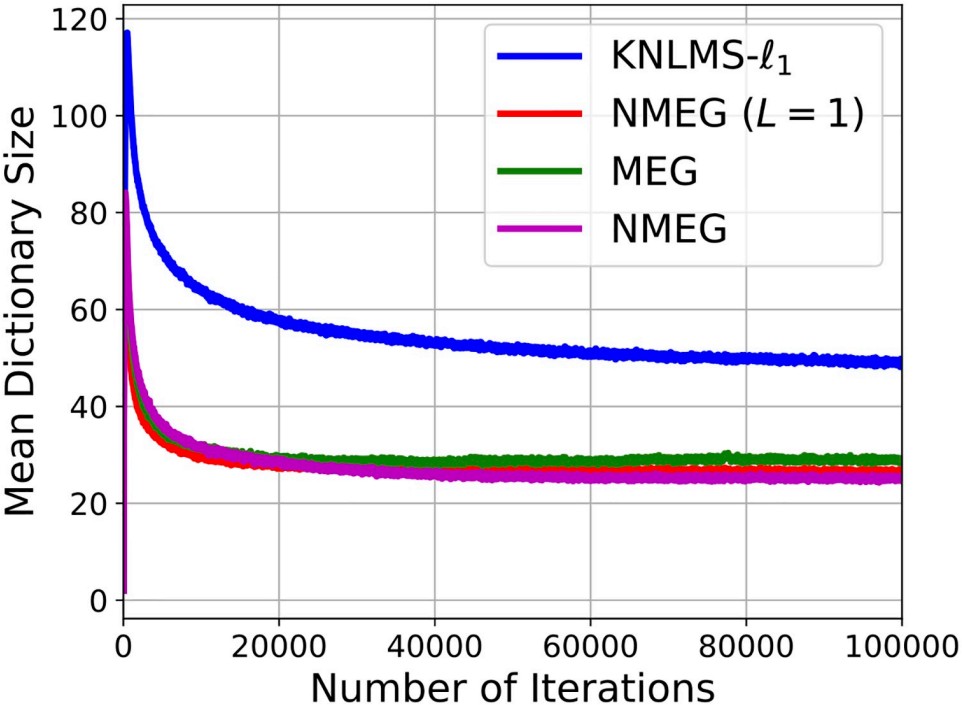

**Fig 4. Dictionary size evolution in experiment described in the Section 4.2.** These results were obtained as averages over 200 independent trials.

matrices $\Gamma$. The NMEG ($L = 1$) converges faster than the other algorithms in this comparison. However, when the iteration index $n$ reaches about 100, 000, the NMEG ($L = 1$) and NMEG exhibit almost the same mean MSE even though the NMEG uses generalized Gaussian kernels. The Fig 4 shows that the NMEG ($L = 1$), the MEG, and the NMEG are able to keep a small dictionary size.

## 4.2 Time series prediction in a toy signal model constructed by generalized Gaussian functions

Further, consider the following synthetic signal model:

$$d^{(n)} := \; 10 \exp\left(-(\boldsymbol{u}^{(n)} - [3,3]^\top)^\top \Lambda (\boldsymbol{u}^{(n)} - [3,3]^\top)\right) + \\ 10 \exp\left(-(\boldsymbol{u}^{(n)} - [7,7]^\top)^\top \Lambda (\boldsymbol{u}^{(n)} - [7,7]^\top)\right), \tag{37}$$

corrupted by the same kind of noise, and driven by the same input sequence, as in the previous experiment. Parameters values pertaining to learning schemes utilized in this experiment are given in Table 3. In addition, the parameters values for the forward-backward splitting scheme are $\lambda = 0.09$ and $\sigma = 0.03$.

We tested the behavior of the proposed adaptive kernel filter theory on two different cases characterized by two instances of $\boldsymbol{\Lambda}$:

$$\begin{pmatrix} 5 & 0.5 \\ 0.5 & 0.2 \end{pmatrix}, \begin{pmatrix} 5 & 0.5 \\ 0.5 & 10 \end{pmatrix}, \tag{38}$$

which have, as smallest eigenvalues, 0.148 and 4.95, respectively. The Fig 5 shows the mean MSE and mean dictionary size of filters at each iteration. In Fig 5a and 5b, the MEG and the NMEG show lower MSE than the KNLMS-$\ell_1$ and NMEG ($L = 1$).

The obtained results confirm the efficacy of using (adaptive) generalized Gaussian kernels. Comparing the MSE curves of the MEG and of the NMEG, it is immediate to see how the performance of the MEG algorithm degrades when the matrix $\boldsymbol{\Lambda}$ is close to singularity, namely when $\Lambda = \begin{pmatrix} 5 & 0.5 \\ 0.5 & 0.2 \end{pmatrix}$, which implies that the term $\log \Gamma_j^{(n)}$ in (20) is difficult to compute, while the NMEG is able to perform well in both cases. The Fig 5c and 5d confirm that the NMEG produces the smallest dictionary. The above results clearly confirm the efficacy of the proposed normalization method for updating precision matrices.

**Table 3. Values of learning parameters in the experiment described in Subection 4.2.**

| Learning algorithm | Parameters values |
|---|---|
| KNLMS-$\ell_1$ | $\rho = 0.03, \gamma = 1.0, \mu = 1.0 \times 10^{-3}, \beta = 0.1$ |
| NMEG ($L = 1$) | $\rho = 0.03, \gamma_{\text{init}} = 1.0, \mu = 1.0 \times 10^{-3}$ $\beta = 0.1, \eta_c = 1.0 \times 10^{-3}, \eta_w = 0.05$ |
| MEG | $\rho = 0.03, \Gamma_{\text{init}} = \boldsymbol{I}, \mu = 1.0 \times 10^{-3}$ $\beta = 0.1, \eta_c = 1.0 \times 10^{-3}, \eta_w = 0.05, L = 2$ |
| NMEG | $\rho = 0.03, \Gamma_{\text{init}} = \boldsymbol{I}, \mu = 1.0 \times 10^{-3}$ $\beta = 0.1, \eta_c = 1.0 \times 10^{-3}, \eta_w = 0.05, L = 2$ |

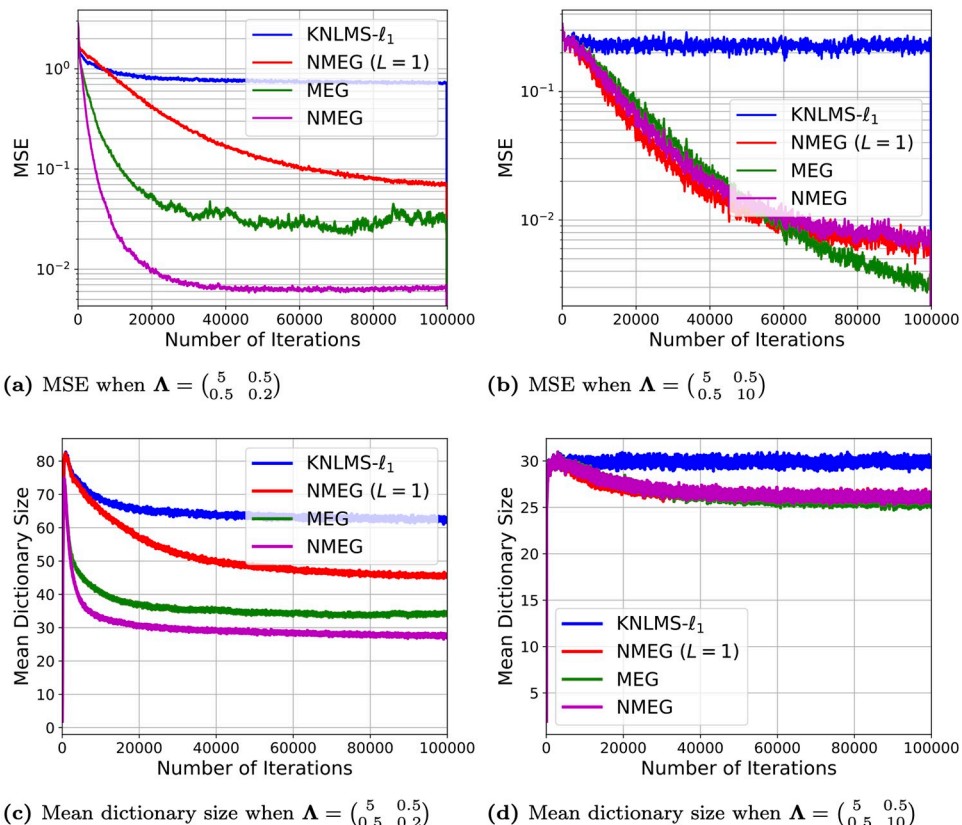

**(a)** MSE when $\boldsymbol{\Lambda} = \begin{pmatrix} 5 & 0.5 \\ 0.5 & 0.2 \end{pmatrix}$ **(b)** MSE when $\boldsymbol{\Lambda} = \begin{pmatrix} 5 & 0.5 \\ 0.5 & 10 \end{pmatrix}$

**(c)** Mean dictionary size when $\boldsymbol{\Lambda} = \begin{pmatrix} 5 & 0.5 \\ 0.5 & 0.2 \end{pmatrix}$ **(d)** Mean dictionary size when $\boldsymbol{\Lambda} = \begin{pmatrix} 5 & 0.5 \\ 0.5 & 10 \end{pmatrix}$

**Fig 5. Performance comparison in experiment described in the Section 4.1.** The learning curves of MSE ((a) and (b)) and mean dictionary size ((c) and (d)) for two different matrices $\boldsymbol{\Lambda}$. These results were obtained as averages over 200 independent trials.

## 4.3 Modeling of a Lorenz chaotic system

Adaptive kernel filters are widely used in time-series prediction [46]. We tested the devised algorithm to model a Lorentz chaotic system [30]:

$$\begin{cases} \frac{dx}{dt} = & -\alpha x + yz \\[6pt] \frac{dy}{dt} = & -\delta(y - z) \\[6pt] \frac{dz}{dt} = & -xy + \theta y - z, \end{cases} \tag{39}$$

where $\alpha = 8/3$, $\delta = 10$, and $\theta = 28$ [11]. The continuous-time equations were sampled by subdividing each unitary interval in 100 sub-intervals. The $x$ component was used to test the algorithm's prediction ability. The $x$ time series was normalized to zero-mean and unit variance. A segment of such time series is displayed in Fig 6.

The input signal to the modeling algorithm was constructed as $\boldsymbol{u}^{(n)} = [x^{(n-5)}, x^{(n-4)}, \ldots, x^{(n-1)}]^{\top}$ and the current value $x^{(n)}$ was taken as the desired response. The values of the learning parameters in this experiment are given in the Table 4. In addition, the parameters for the forward-backward splitting scheme are $\lambda = 0.5$ and $\sigma = 0.05$.

The Figs 7 and 8 show the MSE and the mean dictionary size at each iteration, respectively.

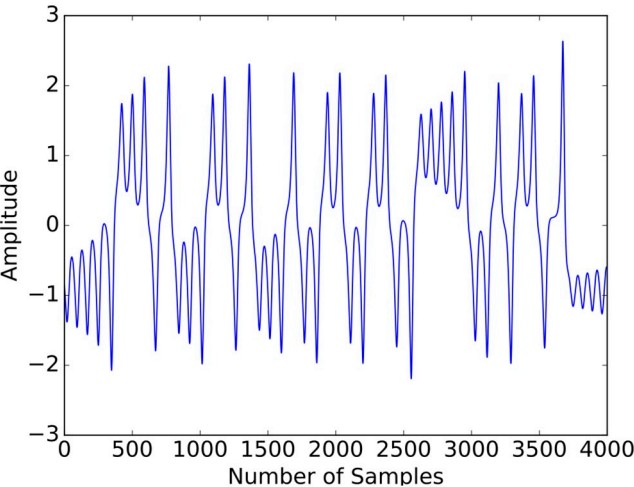

**Fig 6. Segment of the processed Lorenz time series (*x*-component of the flow of the system (39)).**

**Table 4. Values of the parameters in the experiment explained in Section 4.3.**

| Learning algorithm | Parameters values |
|---|---|
| KNLMS-$\ell_1$ | $\mu = 0.5, \rho = 0.05, \gamma = 1.0, \beta = 0.1$ |
| NMEG ($L = 1$) | $\mu = 0.5, \rho = 0.05, \gamma_{\text{init}} = 1.0$ <br> $\beta = 0.1, \eta_c = 0.5, \eta_w = 0.1$ |
| MEG | $\mu = 0.5, \rho = 0.05, \Gamma_{\text{init}} = \boldsymbol{I}$ <br> $\beta = 0.1, \eta_c = 0.5, \eta_w = 0.1, L = 5$ |
| NMEG | $\mu = 0.5, \rho = 0.05, \Gamma_{\text{init}} = \boldsymbol{I}$ <br> $\beta = 0.1, \eta_c = 0.5, \eta_w = 0.1, L = 5$ |

Simulation results indicate that the proposed MEG and NMEG exhibit much better performances, namely, they achieve much smaller mean dictionary size and much smaller MSE values than the other algorithms used for comparison. Comparing the MEG algorithm with the NMEG, the NMEG exhibits better performance in terms of both MSE and mean dictionary size although their parameters are set to the same values. The Fig 9 shows a result of short-term prediction of the Lorenz time series. It can be seen that the NMEG has higher tracking ability than the NMEG ($L = 1$). This result confirms the validity of the proposed model in the case that the components of the input signals are mutually correlated.

## 5 Conclusions

This paper proposed a flexible dictionary learning strategy in the context of generalized Gaussian kernel adaptive filtering, where the kernel parameters are all adaptive and data driven. We introduced a novel update rule for precision matrices, which allows one to update each precision matrix stably thanks to an effective normalization. The main advantage of the proposed approach is that the number of parameters in the proposed generalized Gaussian kernels is larger than the number of parameters in the conventional kernel functions. The adaptation rule of kernel parameters are successfully established within a Lie-group theoretic setting. In addition, together with the $\ell_1$ regularized least squares, the overall kernel adaptive filtering algorithms can avoid overfitting and monotonic inflation of a dictionary. Numerical tests

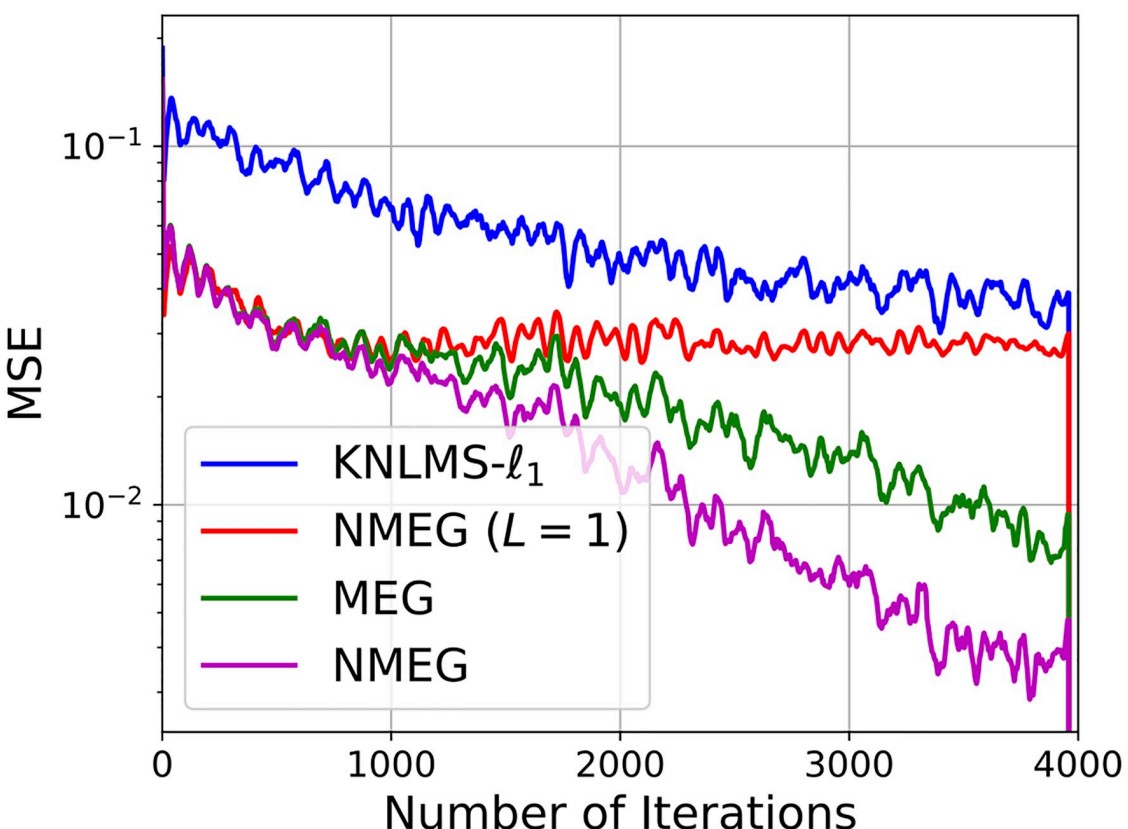

**Fig 7. Convergence curves in the experiment described in the Section 4.3.** These results were obtained as the average over 200 independent trials with different segments of the Lorenz time series.

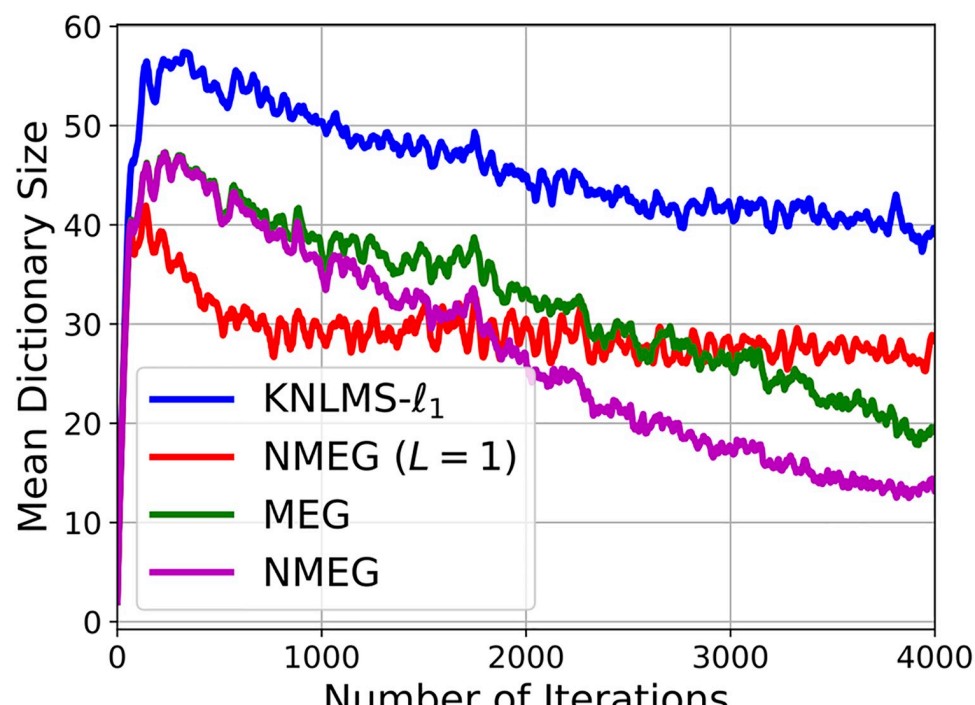

**Fig 8. Dictionary size evolution in the experiment described in the Section 4.3.** These results were obtained as the average over 200 independent trials with different segments of the Lorenz time series.

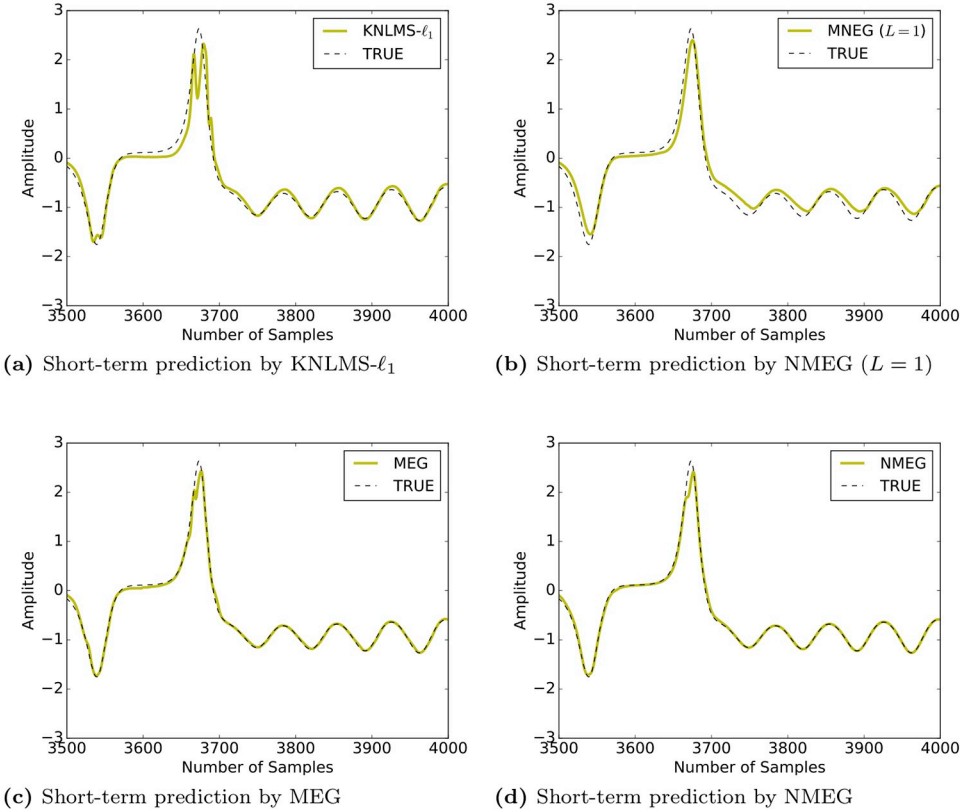

(a) Short-term prediction by KNLMS-$\ell_1$          (b) Short-term prediction by NMEG ($L = 1$)

(c) Short-term prediction by MEG          (d) Short-term prediction by NMEG

**Fig 9. Prediction of the state of a Lorenz chaotic system.** (Plots of the last 500 samples).

confirmed that the proposed algorithm entails lesser mean squared error and dictionary size in modeling nonlinear systems.

## Acknowledgments

This work is supported by JSPS KAKENHI Grant Number 17H01760 and National Center for Theoretical Sciences (NCTS), Taiwan, through a 2016 "Research in Pairs" program.

## Author Contributions

**Conceptualization:** Toshihisa Tanaka.

**Formal analysis:** Simone Fiori.

**Funding acquisition:** Toshihisa Tanaka.

**Investigation:** Tomoya Wada.

**Methodology:** Toshihisa Tanaka.

**Software:** Tomoya Wada, Kosuke Fukumori.

**Supervision:** Toshihisa Tanaka, Simone Fiori.

**Writing – original draft:** Tomoya Wada, Kosuke Fukumori.

**Writing – review & editing:** Kosuke Fukumori, Toshihisa Tanaka, Simone Fiori.

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
