## [Decision Letter · Decision Letter 0]

13 May 2020

PONE-D-20-06282

Generalized Gaussian kernel adaptive filtering

PLOS ONE

Dear Dr. Tanaka,

Thank you for submitting your manuscript to PLOS ONE. After careful consideration, we feel that it has merit but does not fully meet PLOS ONE’s publication criteria as it currently stands. Therefore, we invite you to submit a revised version of the manuscript that addresses the points raised during the review process.

Please revise the paper by considering the reviewer's comments.

We would appreciate receiving your revised manuscript by Jun 27 2020 11:59PM. To enhance the reproducibility of your results, we recommend that if applicable you deposit your laboratory protocols in protocols.io, where a protocol can be assigned its own identifier (DOI) such that it can be cited independently in the future. For instructions see: http://journals.plos.org/plosone/s/submission-guidelines#loc-laboratory-protocols

We look forward to receiving your revised manuscript.

Kind regards,

Jie Zhang

Academic Editor

PLOS ONE

Reviewers' comments:

Reviewer's Responses to Questions

**Comments to the Author**

1. Is the manuscript technically sound, and do the data support the conclusions?

Reviewer #1: Yes

2. Has the statistical analysis been performed appropriately and rigorously? 

Reviewer #1: No

3. Have the authors made all data underlying the findings in their manuscript fully available?

Reviewer #1: Yes

4. Is the manuscript presented in an intelligible fashion and written in standard English?

Reviewer #1: Yes

5. Review Comments to the Author

Reviewer #1: Review Comments

Manuscript number: PONE-D-20-06282

Title Generalized Gaussian kernel adaptive filtering

In this work, authors propose a novel kernel adaptive filtering algorithm, where each Gaussian kernel is parameterized by a center vector and a symmetric positive definite (SPD) precision matrix, which is regarded as a generalization of scalar width parameter. These parameters are adapted on the basis of a proposed least-square type rule to minimize the filtering error. The main contribution of this paper is to establish update rules for precision matrices on the SPD manifold in order to keep their symmetric positive-definiteness. Different from conventional kernel adaptive systems, the proposed filter is a superposition of non-isotropic Gaussian kernels, whose nonisotropy makes the filter more flexible. The kernel adaptive filtering algorithm is established together with an l 1 -type regularization criterion to avoid overfitting and to prevent the increase of dimensionality of the dictionary. Experimental results confirm the validity of the proposed method. Generally, domain of investigation is looks promising and contributions has merits but following major revisions should be incorporated before considering the publication of the manuscript.

1. In the title “Generalized Gaussian kernel adaptive filtering” no any reflection of the novelty and clarity of the contribution. Please change if possible.

2. The qualitative and quantitative advantages and limitations of the proposed scheme should be listed in the abstract.

3. Nomenclature/abbreviations table in the body of the manuscript is required.

4. Quality of English needs extensive re-work, so more attention is required to improve linguistics.

5. Introduction section needs extensive re-work for clarity of the readers; It is suggested to segment the introduction as follows

(1 Introduction, 1.1 related work, 1.2 Innovative contribution, 1.3 organization.

6. Literature review regarding the problem is appropriate but the role/applications/significance of fractional adaptive signal processing algorithm [r1-r5] should be provided qualitatively or quantitative as these modern filtering algorithms outperform their counterparts in term of accuracy and convergence. In active noise control systems as well.

[r1] 2019. A novel application of kernel adaptive filteing algorithms for attenuation of noise interferences. Neural Computing and Applications, 31(12), pp.9221-9240.

[r2] 2019. Design of momentum fractional LMS for Hammerstein nonlinear system identification with application to electrically stimulated muscle model. The European Physical Journal Plus, 134(8), p.407.

[r3] 2019. A new computing paradigm for the optimization of parameters in adaptive beamforming using fractional processing. The European Physical Journal Plus, 134(6), p.275.

[r4] 2019. Normalized fractional adaptive methods for nonlinear control autoregressive systems. Applied Mathematical Modelling, 66, pp.457-471.

[r5] 2015. A new adaptive strategy to improve online secondary path modeling in active noise control systems using fractional signal processing approach. Signal Processing, 107, pp.433-443.7.

7. Please provide a pseudocode of the proposed methodology in elaborative manner, i.e., statements and equations for input, output and intermediate steps.

8. Results and discussion section is too brief. Please include the elaborative description of the results and provide statistical analysis of the results.

9. Advantages and limitations of scheme should also be provided in an elaborative manner in the conclusion section.

6. PLOS authors have the option to publish the peer review history of their article (what does this mean?). If published, this will include your full peer review and any attached files.

Reviewer #1: No

---

## [Author Response · Author response to Decision Letter 0]

5 Jul 2020

Reviewer ===================================

Q1. In the title “Generalized Gaussian kernel adaptive filtering” no any reflection of the novelty and clarity of the contribution. Please change if possible.

R1. We have updated the title to reflect the content of the manuscript.

Q2. The qualitative and quantitative advantages and limitations of the proposed scheme should be listed in the abstract.

R2. The advantage is that we can search for parameters in a wider parameter space that is characterized by the generalized Gaussian function. This generalization brings the need of special treatment of parameters that have a geometric structure called the Lie group.

Q3. Nomenclature/abbreviations table in the body of the manuscript is required.

R3. A table of abbreviations has been added to Section 1.3.

Q4. Quality of English needs extensive re-work, so more attention is required to improve linguistics.

R4. The literal presentation as well as the formulas have been carefully checked for typos and grammatical error.

Q5. Introduction section needs extensive re-work for clarity of the readers; It is suggested to segment the introduction as follows: 1 Introduction, 1.1 related work, 1.2 Innovative contribution, 1.3 organization.

R5. We have segmented the introduction as suggested. Moreover, we have re-organized the existing material and added more details, in particular in Subsection 1.2, regarding the main contributions of the present paper.

Q6. Literature review regarding the problem is appropriate but the role/applications/significance of fractional adaptive signal processing algorithm [r1-r5] should be provided qualitatively or quantitative as these modern filtering algorithms outperform their counterparts in term of accuracy and convergence. In active noise control systems as well.

R6. We have added the suggested references in Subsection 1.1.

Q7. Please provide a pseudocode of the proposed methodology in elaborative manner, i.e., statements and equations for input, output and intermediate steps.

R7. The proposed scheme is summarized through Algorithm 1 in Section 3.4.

Q8. Results and discussion section is too brief. Please include the elaborative description of the results and provide statistical analysis of the results.

R8. We already did a statistical analysis in Subection 4.1, Subection 4.2 and Subection 4.3. To increase the reliability of the analysis we performed 200 independent trials in all the experiments (for example in references [7] and [15] the authors used 50 or 100 independent trials).

Q9. Advantages and limitations of scheme should also be provided in an elaborative manner in the conclusion section.

R9. We have updated the conclusion section according to your suggestion.

---

## [Decision Letter · Decision Letter 1]

31 Jul 2020

Anisotropic Gaussian kernel adaptive filtering by Lie-group dictionary learning

PONE-D-20-06282R1

Dear Dr. Tanaka,

We’re pleased to inform you that your manuscript has been judged scientifically suitable for publication and will be formally accepted for publication once it meets all outstanding technical requirements.

Kind regards,

Jie Zhang

Academic Editor

PLOS ONE

Additional Editor Comments (optional):

Reviewers' comments:

Reviewer's Responses to Questions

**Comments to the Author**

1. If the authors have adequately addressed your comments raised in a previous round of review and you feel that this manuscript is now acceptable for publication, you may indicate that here to bypass the “Comments to the Author” section, enter your conflict of interest statement in the “Confidential to Editor” section, and submit your "Accept" recommendation.

Reviewer #1: All comments have been addressed

2. Is the manuscript technically sound, and do the data support the conclusions?

Reviewer #1: Yes

3. Has the statistical analysis been performed appropriately and rigorously? 

Reviewer #1: Yes

4. Have the authors made all data underlying the findings in their manuscript fully available?

Reviewer #1: Yes

5. Is the manuscript presented in an intelligible fashion and written in standard English?

Reviewer #1: Yes

6. Review Comments to the Author

Reviewer #1: authors have adequately addressed your comments raised in a previous round of review and you feel that this manuscript is now acceptable for publication, No further comments

7. PLOS authors have the option to publish the peer review history of their article (what does this mean?). If published, this will include your full peer review and any attached files.

Reviewer #1: No

---

## [Editor Report · Acceptance letter]

6 Aug 2020

PONE-D-20-06282R1

Anisotropic Gaussian kernel adaptive filtering by Lie-group dictionary learning

Dear Dr. Tanaka:

I'm pleased to inform you that your manuscript has been deemed suitable for publication in PLOS ONE. Congratulations! Your manuscript is now with our production department.

Kind regards,

on behalf of

Dr. Jie Zhang

Academic Editor

PLOS ONE